# Application of NLP Techniques for Detecting Movements Related to Disinformation Strategies on Social Media

Anonymous Full Paper
Submission 42

## Abstract

**Natural Language Processing** (NLP) is a branch of artificial intelligence and computational linguistics that enables the analysis of large amounts of data. This document examines the dataset used in "Oppositional Thinking: Conspiracy Theories vs Critical Thinking Narratives", applying pretrained language models based on the Transformer architecture, such as BERT, RoBERTa, and mDeBERTa. Through these approaches, linguistic and discursive patterns will be identified that characterize each category, with the aim of developing models capable of predicting new content. The results of the study demonstrate the feasibility of using multilingual models to address issues related to disinformation.

## 1 Introduction

In the last decade, social media has generated an exponential increase in data, often serving as a medium for the dissemination of information, which has facilitated the spread of misleading content. Through Noain's study [1] , it was observed that approximately 89.39% of informative content on the internet consisted of hoaxes. Given this scenario, the need arises to develop tools capable of detecting this type of content. **Natural Language Processing** (NLP) makes it possible to analyze large amounts of data. This approach enables the analysis and understanding of text, identifying both linguistic and emotional patterns. The incorporation of machine learning techniques and advanced NLP methods allows for the identification of disinformation. Models such as **BERT** (`Bidirectional Encoder Representations from Transformers`) facilitate text understanding and generation in tasks such as classification and information extraction [2].

## 2 General Framework of Disinformation

**Fake news** are one of the main problems today, as new platforms have enabled faster and easier dissemination.

**Conspiracy theories** are claims shared by one or more individuals or organizations, based on unverified information, with the aim of producing a harmful effect on society.

One of the greatest impacts generated by disinformation is polarization. People generally tend to believe what confirms their own beliefs and values. Although polarization is one of the most significant repercussions, discrimination and institutional trust are also noteworthy.

## 3 Techniques for Disinformation Detection

Due to the exponential growth of information shared on social media, it is necessary to ensure the truthfulness and reliability of the content consumed by users. This has encouraged the introduction of artificial intelligence and natural language processing (NLP) techniques for the identification of patterns within such discourses. Among the most relevant approaches, we can mention the following

**Classical text analysis methods** Mainly based on statistical text representations such as the Bag of Words model or the calculation of term frequency using TF-IDF (`Term Frequency { Inverse Document Frequency`)

**Neural network–based approaches** Neural networks are a computational model inspired by the structure and functioning of the human brain, designed to generate a single data output. These models allow the identification of patterns in text structure and show improvements in the classification of disinformation [3].

**Transformer-based language models** Transformers were first introduced in 2017 through the paper "Attention Is All You Need" by Vaswani et al. They represent a type of neural network architecture that transforms an input sequence into an output sequence by learning from context and tracking sequence components [4]. Among the different transformer-based models, **BERT** (`Bidirectional Encoder Representations from Transformers`), developed by Google, is designed to understand the full context of a word within a sentence.

NLP techniques, in the context of disinformation, combine **computational linguistic methods with machine learning algorithms** to obtain lexical, syntactic, or discursive structures that may indicate intentional manipulation [5].

# 4  Advanced Language Models

**Advanced language models (LLMs)** are deep learning models trained on large amounts of data, capable of understanding and generating language in order to predict the next word in a text sequence and to capture its relationships [6].

**BERT** (Bidirectional Encoder Representations from Transformers) BERT is a language model designed to pretrain deep bidirectional representations from unlabeled text. This model allows us to capture the full context of a word and is based on two main tasks: **Masked Language Modeling (MLM)** and **Next Sentence Prediction (NSP)**. **MLM** is a technique in which a series of tokens in the input sequence are temporarily replaced, and the model is trained to predict the original tokens based on the surrounding context [7]. On the other hand, the **NSP** technique enables the BERT model to learn sentence coherence and sequentiality. Its functioning is based on receiving pairs of sentences and predicting whether the second sentence logically follows the first, with the objective of improving discourse understanding and coherence.

**RoBERTa** (Robustly Optimized BERT Approach) RoBERTa is a variant of the BERT model proposed by Liu et al. (2019), which introduces several improvements in the pretraining phase. The key difference from the BERT model is that it removes the NSP task and employs **dynamic masking**, which generates new random positions at each training epoch. This increases the amount of training data and strengthens the model's ability to predict complex language patterns.

**DeBERTa** (Decoding { enchanced BERT with Disentagled Attention) The DeBERTa model represents a significant improvement over the previously mentioned architectures (BERT and RoBERTa). This model primarily relies on two techniques: the introduction of a disentangled attention mechanism, where each word is represented by two vectors encoding its content and position; and the computation of attention weights between words through disentangled matrices based on their contents and relative positions.

# 5  Methodology and Results

The models used for text classification were **BERT, RoBERTa, and mDeBERTa**, which allow for processing information in multiple languages. The objective is to compare which model provides the best results using the same dataset under similar conditions.

Before training the model, a series of data preprocessing steps were carried out. Since the files were divided by language and by set (training and test),
they had to be loaded individually. Once the datasets were loaded, they were combined, and a basic text cleaning process was performed, which involved removing URLs, mentions, and special characters, as well as trimming leading and trailing spaces when present.

After cleaning the text, label encoding was applied, transforming the categorical labels into integer values (0 and 1) to train the classification model. Following these preprocessing steps, tokenization was performed, in which the input text was broken down into smaller units (tokens), which were subsequently converted into input vectors [8].

The BERT and RoBERTa models were trained in two different scenarios, where the only variations were the number of epochs and the batch size, yielding the following results for each model.

| BERT | | | | XML-RoBERTa | | | |
|---|---|---|---|---|---|---|---|
| Epochs | Training Loss | Validation Loss | Accuracy | Epochs | Training Loss | Validation Loss | Accuracy |
| CASE 1 | | | | | | | |
| 1 | 0.538900 | 0.535585 | 0.768500 | 1 | 0.646400 | 0.650980 | 0.644500 |
| 2 | 0.455700 | 0.498312 | 0.802000 | 2 | 0.641400 | 0.654072 | 0.644500 |
| 3 | 0.363300 | 0.459254 | 0.826000 | 3 | 0.490700 | 0.520073 | 0.755000 |
| CASE 2 | | | | | | | |
| 1 | 0.503300 | 0.544836 | 0.766000 | 1 | 0.638200 | 0.620271 | 0.644500 |
| 2 | 0.557200 | 0.525693 | 0.768500 | 2 | 0.658900 | 0.605679 | 0.708500 |
| 3 | 0.421300 | 0.541324 | 0.796500 | 3 | 0.601800 | 0.642016 | 0.706500 |
| 4 | 0.409200 | 0.471091 | 0.814000 | 4 | 0.570200 | 0.665658 | 0.709000 |
| 5 | 0.391400 | 0.472180 | 0.817000 | 5 | 0.541400 | 0.630709 | 0.717000 |

**Figure 1.** Results of the BERT and RoBERTa Models

In the case of the **BERT model**, the first scenario achieved a final accuracy of 82.6%, whereas the second scenario, despite having more epochs, reached an accuracy of 81.7%. This indicates that the inclusion of a greater number of epochs slightly decreased the model's performance. Additionally, validation loss increased during the last two epochs of the second scenario, suggesting a possible onset of **overfitting**.

For the **XML-RoBERTa model**, a similar pattern to BERT is observed: adding more epochs did not lead to an improvement in model accuracy. In the first scenario, the maximum accuracy (75.5%) was reached in the last epoch, as in the second scenario; however, in the latter, the increase in accuracy per epoch was minimal. Moreover, an increase in validation loss was observed even as model accuracy improved. This behavior suggests that, although the model continues to learn from the training data, its generalization capability on the validation set begins to deteriorate. The simultaneous increase in validation loss and decrease in accuracy (fourth epoch) indicates signs of overfitting.

It can be observed that both models achieve better results when the number of epochs is lower, striking

an optimal balance between learning and the model's generalization capability.

Next, the results from the first scenario of these models are compared with those obtained for the **mDeBERTa model**.

| Modelo | Épocas | Training Loss | Validation Loss | Accuracy |
|---|---|---|---|---|
| BERT multilingüe base | 1 | 0,5389 | 0,5356 | 0,7685 |
| | 2 | 0,4557 | 0,4983 | 0,8020 |
| | 3 | 0,3633 | 0,4593 | 0,8260 |
| XML Roberta base | 1 | 0,646400 | 0,650980 | 0,644500 |
| | 2 | 0,641400 | 0,654072 | 0,644500 |
| | 3 | 0,490700 | 0,520073 | 0,755000 |
| mDeBERTa | 1 | 0,392100 | 0,370994 | 0,858500 |
| | 2 | 0,311100 | 0,440613 | 0,853500 |
| | 3 | 0,196400 | 0,472871 | 0,883000 |

**Figure 2.** Comparations

The three models analyzed exhibit differentiated performance depending on the number of epochs, reflecting dynamics that allow for discussion of their generalization capability and the potential onset of **overfitting**.

The mDeBERTa model shows superior performance from the first epoch compared to the other models (`85.8% accuracy`), demonstrating rapid adaptation to the dataset. However, despite reaching its maximum accuracy in the last epoch (`88.3%`), there are signs of potential overfitting. While the training loss continues to decrease, the validation loss begins to increase from the second epoch, indicating a loss in generalization capability.

In conclusion, the **BERT model** shows a more solid and stable performance, achieving adequate results that are not influenced by overfitting. In contrast, the **XML-RoBERTa model**, despite being designed for multilingual tasks, seems less effective at capturing the relevant patterns.

In this study, the **mDeBERTa model** delivers the best performance in terms of accuracy, outperforming the other models from the first epoch of training, although there is a noticeable risk of early-stage overfitting.

# 6 Current Strategies for Disinformation Detection

The detection of disinformation has evolved significantly in recent years thanks to artificial intelligence.

Early approaches focused on natural language processing (`NLP`) techniques and supervised learning models. In contrast, more recent developments concentrate on deep learning models, specifically large language models (`LLMs`) [9, 10].

The increase of disinformation on the internet and its wider dissemination make automated fact-checking essential today. Automated fact-checking is one of the most significant advances in the application of artificial intelligence for disinformation detection. Its primary goal is to evaluate the truthfulness of contextual claims by comparing them with reliable external sources [11].

The introduction of deep learning models and pretrained language models has considerably changed the landscape. One example is FEVER (`Fact Extraction and VERification`), introduced by Thorne et al. (`2018`), which enabled the standardization of large-scale automated fact-checking and fostered the development of more advanced systems.

**Multimodal Analysis** Disinformation can manifest through images and videos generated using deep learning techniques. Therefore, the use of multimodal models is useful for detecting fake news on social media. Multimodal analysis has emerged due to the limitations of text-based approaches, which face difficulties in capturing the complexity of messages that combine multiple sources of information.

Multimodal analysis integrates different modalities to detect inconsistencies or manipulation patterns that help identify disinformative content [9]. One of the most prominent approaches within multimodal analysis is the CLIP model, developed by OpenAI in 2021. This model is trained on large-scale text-image pairs, learning to associate textual descriptions with visual representations.

**Graph-Based Models** Due to the requirements of disinformation detection, graph-based models have been incorporated, which allow for identifying how content spreads across networks.

In this context, Graph Neural Networks (`GNNs`) stand out. They are designed to process graph structures directly, where information is organized into nodes and edges. The fundamental principle of GNNs is local information aggregation, in which each node updates its representation (`embedding`) by combining its own features with those of its neighbors [12].

Therefore, GNNs are particularly useful in disinformation detection, as they allow for representing the diffusion structure in addition to the textual content.

**Explainable Artificial Intelligence (`XAI`)** The use of deep learning models in disinformation detection offers numerous advantages. However, it faces the challenge of lack of interpretability.

The concept of **Explainable Artificial Intelligence (`XAI`)** refers to a set of techniques and methodologies designed to make machine learning systems understandable and interpretable for users [13].

Among the most widely used XAI techniques in **Natural Language Processing (`NLP`)** and disinformation detection are: **LIME** (`Local Interpretable Model-agnostic Explanations`), which allows identifying which text features have most influenced the classification of content

by generating perturbations on the input and observing changes in predictions; and **SHAP** (`Shapley Additive exPlanations`), which assigns contribution values to each input feature, enabling an understanding of its impact on the prediction [14].

# 7 Conclusion

In the evaluation of the dataset from the study "Oppositional Thinking: Conspiracy Theories vs Critical Thinking Narratives", various Transformer models were applied, leading to the conclusion that the choice of the number of training epochs is crucial, as excessive training can result in overfitting or model deterioration. After training each model, a series of accuracies were obtained, with mDeBERTa achieving the best results from the first epoch. However, from the second epoch onwards, signs of overfitting were observed, highlighting the need to apply more advanced regularization techniques.

The introduction of automated fact-checking models and multimodal analysis allows for the evaluation and detection of disinformation beyond the linguistic analysis performed with the various Transformer models, providing insights into how such narratives are generated, disseminated, and evolve. This is complemented by the growing interest in Explainable Artificial Intelligence (`XAI`), which aims to enhance the transparency and interpretability of detection systems.

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
