# OpenReview forum: "Application of NLP Techniques for Detecting Movements Related to Disinformation Strategies on Social Media"
_NLDL.org/2026/Conference — Submitted to NLDL 2026_

### Official Review · Reviewer_2eZ1 · 2025-09-17
**Unclear contribution and weak evaluation**

**Rating:** 1
**Confidence:** 5
**Final Rating:** 1
**Final Confidence:** 5

**Summary:**

In this article, they train three models from the BERT family for a binary classification task, where the categories are "conspiracy theory" and "critical thinking".

**Strengths:**

**S1:** Detecting disinformation on Social media is an important task.

**Weaknesses:**

**W1:** The contribution and methodology of the paper is unclear. The title and abstract does not clearly convey what the article is about.
The title says "disinformation strategies", but models are only trained on a binary classification of "conspiracy" vs "critical thinking".
The dataset is never properly described, I have assumed it was the dataset from [1]. Also, it is not described how the BERT models are made into classifiers.

**W2:** The contribution of the article seems to be evaluation, but only three models are trained on a single dataset.
If evaluation is the only thing you contribute, it will have to be thorough.

**W3:** Only 1-2 seeds are trained for each model and the criteria for stopping training are not clear.
Model performance varies for each retraining. Therefore you should decide on a training procedure and then train several seeds
to get an idea about the average performance of your model types.

Extra comment:
When you reference an article, make sure you make a proper reference. Only citing the title is not enough.


[1] Balasundaram, P., Swaminathan, K., Sampath, O., & Km, P. (2024). Oppositional Thinking Analysis: Conspiracy Theories vs Critical Thinking Narratives. Working Notes of CLEF.

**Final Justification:**

The article is not clear about what the contribution is meant to be, but it seems to be evaluation of models from the BERT family on the task of distinguishing critical thinking from conspiracy theories. However, the evaluation is very weak with only three models on one dataset and only 1-2 seeds for each model. This is not material for a scientific conference.

**Justification:**

The article is not clear about what the contribution is meant to be, but it seems to be evaluation of models from the BERT family on the task of distinguishing critical thinking from conspiracy theories.
However, the evaluation is very weak with only three models on one dataset and only 1-2 seeds for each model.
This is not material for a scientific conference.

---

### Official Review · Reviewer_PYCm · 2025-10-08
**Review of "Application of NLP Techniques for Detecting Movements Related to Disinformation Strategies on Social Media"**

**Rating:** 2
**Confidence:** 4

**Summary:**

This submission investigates the use of Transformer-based language models, specifically BERT, RoBERTa, and mDeBERTa, for classifying social-media content into disinformation-related categories. Using the dataset from “Oppositional Thinking: Conspiracy Theories vs Critical Thinking Narratives”, the authors compare model performance under different epoch and batch size settings.

The paper reports that mDeBERTa achieves the best accuracy (~88.3%) but shows early overfitting. The authors also discuss multimodal detection, graph-based analysis (GNNs), and Explainable AI (XAI) methods such as LIME and SHAP as complementary approaches.

While the problem addressed is relevant and timely, the current manuscript is not within the expected length (3.5 pages of main text vs. 5-8 required according to the authors instructions) and does not include enough experimental detail, quantitative analysis, or supporting results to qualify as a complete full paper submission.

**Strengths:**

1. The detection of disinformation through NLP and deep learning is of clear social and scientific significance. The paper identifies a real-world problem that aligns with current research trends.

2. The explanation of BERT, RoBERTa, and mDeBERTa is accurate and concise, showing a good understanding of their architectural differences.

3. The discussion of graph neural networks, multimodal analysis, and XAI methods demonstrates awareness of the larger research ecosystem and situates the study within it.

4. Despite its brevity, the manuscript is well-structured, technically consistent, and easy to follow.

**Weaknesses:**

1. Below Minimum Page Requirement: The paper’s main content spans only 3.5 pages, below the 5–8 page limit required for full papers.

2. The study modifies only the number of epochs and batch size. There is no analysis of learning rate schedules, model regularization, or tokenization strategies.

3. Reporting only accuracy obscures potential biases. Class imbalance would render accuracy unreliable; metrics such as macro-F1 and AUC should be provided.

4. The dataset is referenced but not sufficiently described, no indication of size, class ratios, language composition, or preprocessing details beyond cleaning steps.

5. No significance testing is applied, weakening claims that one model “outperforms” another.

6. Although overfitting is identified, no mitigation experiments are reported.

7. The paper’s main novelty lies in the dataset and comparative context rather than the modeling approach itself, which uses standard architectures without methodological innovation.

8. Sections on multimodal detection and XAI are literature summaries, not experiments. They read as appendices rather than evidence-based sections.

**Justification:**

This paper addresses an important societal and technical problem, the automatic detection of disinformation using Transformer models. It shows conceptual competence and a clear understanding of model architecture and generalization dynamics. However, the manuscript falls short of NLDL full-paper standards due to its length (< 5 pages), lack of methodological depth, and minimal evaluation.

While the findings suggest that mDeBERTa performs best for this dataset, the empirical evidence is insufficient to substantiate the claim. The paper could serve as a short paper or work-in-progress submission, but it is not ready for publication in the NLDL 2026 proceedings in its current form.

---

### Official Review · Reviewer_3Uhf · 2025-10-09
**Not Enough**

**Rating:** 1
**Confidence:** 4

**Summary:**

This paper considers the task of detecting disinformation. The paper does a reasonable job stating why the problem is important and doing a quick review of techniques used. The papers trains different types of transformers and reports their results

**Strengths:**

The paper is on an important topic and the related work section is well written. However, I think the paper has several experimental shortfalls.

**Weaknesses:**

There are many weaknesses.

1. Dataset. - There is no description of the data used. Where is it sourced, how many data points are there, or even what the data looks like. This is something that is very important that is missing.

2. Training details. The concrete details of training the models is missing. For example Table 1 says case 1 and case 2 and it is unclear what the difference between these cases are.

3. The experimental framework. Only 3 models are considered and they are trained for very few epochs. It could be the case that is reasonable because of the size of the data, however, without knowing details about the data this is not feasible to tell.

4. Other approaches. The paper ends with section 6, which talks about more approaches. While this is interesting, it is again more about prior work, and is not directly related to any experiment in the paper.

**Justification:**

The paper is runs a single experiment, for which many details are missing.

---

### Official Review · Reviewer_icgZ · 2025-10-10

**Rating:** 1
**Confidence:** 5
**Final Rating:** 1
**Final Confidence:** 5

**Summary:**

The authors aim to study the spread of disinformation and present some experiments with BERT and derivatives.

**Strengths:**

Disinformation and fake-news are probably the greatest virtual threat. We urgently need to fight against false news by all means and in that respect the paper is timely and aiming at the right direction.

**Weaknesses:**

The scientific approach and outcome of the paper is not significant. Firstly, the deployed models are rather old, there is only one data set, the take-home message deals with overfitting but appears unrelated to the topic of the paper. The conclusion is that more sophisticated regularization techniques need to be incorporated and I wonder why this hasn‘t been done? The discussed strategies (e.g., graph networks,…) are also not tested or incorporated. There is not enough in it for publication yet.

Some references cited in the text don‘t show up in the reference section.

**Final Justification:**

There was no author response. See comments in review.

**Justification:**

In sum, novelty is low, the outcomes are well known and the connection to disinformation is not clear.

---

### Meta-Review · Area_Chair_bKHT · 2025-11-01

**Recommendation:** Reject
**Confidence:** 4

**Metareview:**

The paper addresses the problem of disinformation in social media.

The quality of the presentation and content is not sufficient for the NLDL conference due to its length (less than 5 pages), lack of methodological depth, and inadequate evaluation.

pros:
1. The paper addresses a theme that could be of interest
2. The authors use language models to address this problem

cons:
1. Lack of description of the data
2. limited experimentation framework
3. length of the manuscript
4. organization of the content
5. Use of old models


I would therefore suggest to reject of the paper in its current state.

---

### Decision · Program_Chairs · 2025-11-05

**Decision:**

Reject

**Comment:**

Based on the reviewers and AC comments, the paper cannot be presented at the conference.